# Association of *IRS1* (Gly972Arg) and *IRS2* (Gly1057Asp) genes polymorphisms with OSA and NAFLD in Asian Indians

**Surya Prakash Bhatt** [ID]*, **Randeep Guleria**

Department of Pulmonary, Critical Care and Sleep Medicine, All India Institute of Medical Sciences, New Delhi, India

* suryabhat@gmail.com

**Data Availability Statement:** Complete data has been incorporated in the manuscript

**Funding:** Funding agency: Department of Science and Technology, Ministry of Science and

## Abstract

### Aim and objective

The aim of the study was to investigate the relationships between insulin receptor substrate (*IRS*) 1 (Gly972Arg) and *IRS2* (Gly1057Asp) genes with obstructive sleep apnea (OSA) and non-alcoholic fatty liver disease (NAFLD) in Asian Indians.

### Method

A total of 410 overweight/obese subjects (130 with OSA with NAFLD, 100 with OSA without NAFLD, 95 without OSA and with NAFLD and 85 without OSA and without NAFLD) were recruited. Degree of NAFLD was based on liver ultrasound and of OSA on overnight polysomnography. Genotyping was performed by polymerase chain reaction-restriction fragment length polymorphism and confirmed by gene sequencing.

### Result

Mean values of blood pressure, body fat markers, blood glucose, lipids, liver function, and markers of insulin resistance were significantly increased in OSA and NAFLD subjects (p<0.05). In addition, according to age (years) categories, blood pressure, blood glucose, lipids, obesity markers, and markers of insulin resistance were significantly higher in 45–60 years group as compared to 20–45 years group (p<0.05). In *IRS1* gene, the genotype frequency (%) of Arg/Arg was significantly higher in NAFLD and OSA subjects. In addition, Gly/Arg genotype of *IRS1* gene was associated with significantly higher body mass index, fat mass, %body fat, triglycerides, cholesterol, alkaline phosphate, aspartate transaminase, fasting insulin and HOMA-IR levels in OSA and NAFLD subjects. No significant difference in genotype frequencies of *IRS2* was observed between four groups. Further we found that subjects carrying *IRS1* Gly/Arg (OR 4.49, 95% C.I. 1.06–12.52, p = 0.002) genotype possess a much higher risk of OSA and NAFLD compared to *IRS2* Gly/Asp (OR 1.01, 95% C.I. 0.8–2.56, p = 0.05). In sub group analysis of *IRS1* Gly/Arg have significant differences between the mild, moderate and severe group (P<0.05). In addition, patients with the 'Gly' allele were inclined to develop more severe OSA.

Technology, SR/SO/HS-0146/2010, Dr. Surya Prakash Bhatt.

**Competing interests:** The authors have declared that no competing interests exist.

**Abbreviations:** OSA, Obstructive sleep apnea; NAFLD, non-alcoholic fatty liver disease; T2DM, type 2 diabetes mellitus; CVD, cardio vascular disease; WC, waist circumference; HC, hip circumference; MTC, mid-thigh circumference; FBG, fasting blood glucose; TC, total cholesterol; TG, triglycerides; HDL-C, high-density lipoprotein cholesterol; LDL-C, low-density lipoprotein cholesterol; AST, aspartate aminotransferase; ALT, alanine aminotransferase; HOMA, homoeostasis modal assessment; PSG, polysomnography; AHI, *apnea–hypopnea index.*

## Conclusion

We concluded that Asian Indian subject carrying the allele Gly972Arg polymorphism of *IRS1* is predisposed to develop OSA and NAFLD.

## Introduction

Obstructive sleep apnea (OSA) is a common sleep problem in which complete airway obstruction, caused by pharyngeal collapse during sleeping time. The global prevalence in general populations is 9–38% [1]. In Indian studies, the prevalence of OSA is 4.4–13.7% [2]. In addition, OSA in Indian males varies from 4.4–19.7% and in females, it is between 2.5–7.4% from the previous studies [2]. The prevalence of OSA also varies depending on the diagnostic criteria used and the age and sex of the population.

Non-alcoholic fatty liver disease (NAFLD) has a broad spectrum from fatty infiltration to severe fibrosis, cirrhosis, and hepatocellular carcinoma. The global prevalence of NAFLD among general population ranged from 11.2% - 37.2%). Similar ranges were shown of biopsy-confirmed nonalcoholic steatohepatitis (NASH) among NAFLD subjects ranging from 15.9% to 68.3% [3]. Prevalence of NAFLD in the Asian population was reported as 31% [4]. In our previous study, we reported that the prevalence of NAFLD is 24.5–32.2% [5] and the primary risk factors for NAFLD are obesity, type 2 diabetes mellitus (T2DM), dyslipidemia, and insulin resistance.

Clinical finding showed that OSA has been associated with NAFLD [6]. Recent Meta-analysis has been reported that OSA was independently associated with NAFLD in terms of liver enzymes and histological alterations [6]. OSA causes accumulation of fatty acids in the liver as a result of nocturnal hypoxia, insulin resistance, metabolic syndrome, dyslipidemia, hypertension, oxidative stress and systemic inflammation. Another study has been indicated that nocturnal hypoxia causes NAFLD development and progression [7]. However, nocturnal hypoxia is correlated with development and progression of NAFLD in OSA patients [8].

Insulin receptor is a hetero tetramer consisting of alpha (α) and beta (β) dimers. The α-subunit consisting of the ligand-binding site, while the β-subunit consists of a ligand-activated tyrosine kinase. On ligand binding, when tyrosine is phosphorylated, the insulin receptor gets converted into two intracellular substrates, insulin receptor substrate (IRS)-1 and insulin receptor substrate (IRS)-2 [9]. The gene for *IRS1* is located on chromosome 2q36 and encodes a 1,242-amino acid protein. The most common polymorphism in the IRS *-1* gene (Gly972Arg), was reported to be associated with OSA [10] and NAFLD [11]. The *IRS2* gene is located on chromosome 13q34 and encodes a protein of 1,354 amino acids. Moreover, the common polymorphism Gly1057Asp in the *IRS2* gene has also been reported to influence the susceptibility to insulin resistance and T2DM in polycystic ovary syndrome women [12, 13]. Till date, no studies have been investigated the association of *IRS1* and *IRS2* gene with OSA and NAFLD in Asian Indians.

We hypothesized that the *IRS1* (Gly972Arg) and *IRS2* (Gly1057Asp) genes may influence insulin resistance and are associated with risk of OSA and NAFLD in overweight non-diabetic Asian Indians. The aim of the present study was to investigate the relationships between *IRS1* and *IRS2* gene polymorphisms with OSA and NAFLD in Asian Indians.

## Methodology

### Subjects

A total of 410 overweight/ obese subjects [body mass index (BMI>23kg/m$^2$)] with age from 20 to 60 years were evaluated from Outpatients Department of Pulmonary, Critical Care and

Sleep Medicine at All India Institute of Medical Sciences (AIIMS), New Delhi, India between July 2012 to July 2018. Out of 410 subjects, 130 with OSA with NAFLD (group 1), 100 with OSA without NAFLD (group 2), 95 without OSA and with NAFLD (group 3) and 85 without OSA and without NAFLD (group 4) subjects have been recruited. The study was approved by the Institutional Ethics Committee of AIIMS, New Delhi, India. All experiments were performed in accordance with relevant guidelines and regulations. Written informed consent was obtained from all participants. Subjects with known T2DM, cardiovascular disease, other liver diseases, severe chronic obstructive pulmonary disease/ advanced lung disease with mechanical upper airway obstruction, severe organ damage, human immunodeficiency virus infection, pregnancy, and lactation, or with any pro-inflammatory state were excluded from the study.

## Clinical, anthropometric and biochemical investigations

Blood pressure was measured over the right arm in sitting position after five-minute rest. Measurement of weight, height, body mass index, waist circumference (WC), hip circumference (HC), mid-thigh circumference (MTC) and skinfold thickness at 6 sites (triceps, biceps, anterior axillary, suprailiac, subscapular and lateral thoracic) were measured according to the methods adopted in the previous study [14]. Investigation of fasting blood sugar (FBS), total cholesterol (TC), serum triglycerides (TG), high-density lipoprotein cholesterol (HDL-C), low-density lipoprotein cholesterol (LDL-C), aspartate aminotransferase (AST) and alanine aminotransferase (ALT) levels were done as previously described [15]. Fasting serum insulin levels were measured by chemiluminescence (inter-assay CV 4.3%) using a Siemens Immulite 2000 (Siemens Healthcare). Hyperinsulinemia was defined by values in the highest quartile [13]. The value of Homoeostasis Model Assessment of insulin resistance (HOMA-IR) was calculated as: fasting insulin (IU/ml) × fasting glucose (mmol/l)/22·5 [16].

## Ultrasound imaging

All subjects were assessed by an abdominal ultrasound using 3.5MHz curvilinear probe (Siemens-G 60 S 2004, Germany). For this entire study, abdominal ultrasound was done by a single radiologist. The definition of fatty liver was based on a comparative assessment of image brightness relative to the kidneys, in line with previously reported diagnostic criteria [17].

## Overnight polysomnography

All subjects were called for the overnight sleep study at 8.00 pm and were attached to Alice 3 infant and adult computerized polysomnography (PSG) system using the various leads and devices through standard gold cup electrodes [18]. Overnight PSG was recorded according to standard protocols [19]. Diagnosis of OSA was made on the basis of international classification of sleep disorders (ASDA, diagnostic classification steering committee). Breathing event was defined according to the commonly used clinical criteria published by American Academy of Sleep Medicine Task Force [18]. PSG was conducted in a single sleep laboratory and analysis was done by a single expert.

**Genetic investigations.** Genomic deoxyribonucleic acid (DNA) was extracted from whole blood using the QIAamp DNA extraction kit (Qiagen, Hilden, Germany) and stored at -20°C for the further experiments. The DNA concentration of the samples was 80 to 90 ng/mL. Genotyping was performed by polymerase chain reaction-restriction fragment length polymorphism (PCR-RFLP). Its concentration and quality were then measured in a Nanodrop (Thermo Scientific, Waltham, MA, USA). DNA amplification and RFLP of the *IRS1* (Gly972Arg) and *IRS2* (Gly1057Asp) genes were performed by previously reported studies [20,

21]. In this study, 60 samples has been confirmed for each polymorphism using DNA sequencing analysis.

## Statistical analysis

Data was entered in an Excel spreadsheet (Microsoft Corp, Washington, USA). The distribution of clinical, biochemical, anthropometric and body composition parameters were confirmed to approximate normality. Categorical data was analyzed by Chi-squared test, with Fisher correction when appropriate, and expressed as absolute number (%). Continuous variables were expressed as the mean ± standard deviation to summarize the variables. All continuous values were performed using the Z score method. The influence of the groups (1vs2, 1 vs3, 1vs 4, 2vs3 and 2vs4) was estimated by the Analysis of Covariance (ANCOVA) test with multiple comparisons. Pearson's correlation coefficient and significance of 'r' were used to compare the inflammatory marker levels and clinical parameters.

In order to determine if observed allele frequency was in conformity with the expected frequency (Hardy Weinberg equilibrium), chi-square analysis was done. Between-group differences in proportions of alleles or genotypes were compared using Chi-square test and a two-tailed Fisher's exact test. The influence of the genotype on the clinical biochemical, anthropometric and body composition parameters was estimated by ANCOVA, whether there are any statistically significant differences between the groups. Logistic regression analyses were carried out to identify the differences in genotypic frequencies and interaction of two SNPs between the groups. Bonferroni corrections for multiple comparisons were performed. The odds ratio (OR) and 95% confidence interval were used as a measure of strength for the association between *IRS1* (Gly972Arg) and *IRS2* (Gly1057Asp) genotypic combinations with the disease. In addition, subgroup analysis was conducted to see the relationship of OSA and gene polymorphisms. A p-value <0.05 was considered as significant.

## Results

### Clinical, body composition, anthropometry and biochemical profiles

Based on age (20–44 years and 45–60 years) category, clinical, body composition, anthropometry and biochemical profiles and detailed multi variable comparison (group 1vs2, 1 vs3, 1vs 4, 2vs3, 2vs4 and 3vs4) are presented in Tables 1–3. It was observed that the mean values of blood pressure (systolic and diastolic) (p<0.05), BMI (p = 0.003), fat mass (p = 0.02) and %body fat (p = 0.002) was significantly higher in OSA with NAFLD group as compared to other groups.

Mean values of WC (p = 0.001), HC (p = 0.003), MTC (p = 0.005), neck circumference (p = 0.0004), suprailiac (p = 0.02), lateral thoracic (p = 0.02) and thigh (p = 0.05) was significantly higher in OSA with NAFLD group as compared to other groups.

The values of FBS (p = 0.004), serum TG (p = 0.02), TC (p = 0.02), HDL (p = 0.005), LDL (p = 0.002), AST (p = 0.01), ALT (p = 0.03), ALP (p = 0.05), fasting Insulin (p = 0.001) and HOMA-IR (p = 0.001) were significantly increased in OSA with NAFLD group.

According to age (years) categories, we found that blood pressure, fasting blood glucose, lipids, obesity markers (BMI, body fat, WC, HC and WHR), fasting insulin and HOMA-IR was significantly higher in 45–60 years group as compared to 20–45 years group (p<0.05).

### Genotype distribution of IRS1 (Gly972Arg) and IRS2 (Gly1057Asp) genes

The group wise genotypic frequencies of *IRS1* (Gly972Arg) SNP are presented in Table 4. Overall, 78.75% of subjects were *Gly/Gly* homozygous, 15.83% were Gly/Arg heterozygous, and 5.42% were Arg/Arg homozygous. Higher frequency of Arg/Arg genotype of *IRS1* gene

**Table 1. Clinical and body composition investigations.**

| Variables | OSA with NAFLD (n = 130) | | OSA without NAFLD (n = 100) | | Without OSA and with NAFLD (n = 95) | | Without OSA and without NAFLD (n = 85) | | Overall P value |
|---|---|---|---|---|---|---|---|---|---|
| | Age (20–44) | Age (45–60) | Age (20–44) | Age (45–60) | Age (20–44) | Age (45–60) | Age (20–44) | Age (45–60) | |
| Systolic blood pressure (mmHg) | 130.4±9.5[¶] | 140.4±10.5[¶] | 128.2 ±14.65[@] | 129.2±15.6[@] | 124.6±18.65 | 125.6±20.1 | 116±14.56 | 125±16.9 | 0.001 |
| Diastolic blood pressure (mmHg) | 80.23±14.4 | 88.65±14.4 | 84.65±12.45 | 86.65±13.6 | 79.56±14.65 | 89.65±15.4 | 75±9.65 | 87.6±12.3 | 0.003 |
| Pulse rate (minutes) | 71.23±6.54 | 79.7±7.8 | 73.6.71±5.87 | 76.54±5.9 | 75.6±5.87 | 76.56±5.1 | 77.25±4.26 | 77.65±4.6 | 0.11 |
| Body mass index (Kg/m$^2$) | 30.23±8.54[¶] | 36.56±7.9[¶] | 30.23±6.57[@] | 32.5±6.9[@] | 29.65±8.67 | 31.0±8.3 | 26.56±5.65 | 28.54±8.6 | 0.003 |
| Fat mass (kg) | 35.65±16.54* | 41.45±17.4* | 35.6±14.65 | 38±14.2 | 31.1±14.56 | 36.1±14 | 30.5±9.54 | 35.5±9.5 | 0.02 |
| Fat free mass (kg) | 54.1±13.54* | 56.1±12.1* | 52.4±10.23[@] | 53.4±11.7[@] | 48.03±11.23 | 47.03±12.1 | 45.7±8.96 | 46.53±9.7 | 0.002 |
| Total body water (kg) | 36±9.65 | 40.6±8.6 | 34.2±10.23 | 38.2±9.6 | 31.2±9.54 | 35.2±8.7 | 30.2±7.91 | 33.2±8.3 | 0.5 |
| Body fat (%) | 36.5±12.54[¶, Y] | 40.2±13.6[¶, Y] | 32.2±14.56 | 38.2±11.6 | 34.1±11.56 | 36.65±12.8 | 34.6±10.2 | 35.6±11.6 | 0.002 |

Results are shown as mean± SD. P value ≤0.05 is statistically significant. One-way analysis of variance (ANOVA) were carried out.

*Group 1 *vs* 2, 1*vs* 3 and 1 *vs* 4 (p≤0.05)

[#] group 3 vs 4 (p≤0.05)

[@] group 2 *vs* 4 (p≤0.05)

[¶] group 1 *vs* 4 (p≤0.05)

[‡] group 2 *vs* 3 (p≤0.05)

[Y] group 1 *vs* 3 (p≤0.05).

**Table 2. Anthropometry parameters.**

| Variables | OSA with NAFLD (n = 130) | | OSA without NAFLD (n = 100) | | Without OSA and with NAFLD (n = 95) | | Without OSA and without NAFLD (n = 85) | | Overall P value |
|---|---|---|---|---|---|---|---|---|---|
| | Age (20–44) | Age (45–60) | Age (20–44) | Age (45–60) | Age (20–44) | Age (45–60) | Age (20–44) | Age (45–60) | |
| **Circumferences (cm)** | | | | | | | | | |
| Waist | 103±12.45[¶, Y] | 106.9±13.3[¶, Y] | 100±13.87 | 104.2±14.6 | 98±12.89 | 102±13.5 | 95±15.4 | 100±15.6 | 0.001 |
| Hip | 104±13.14* | 109.5±13.2* | 102±22.35 | 106.5±23.5 | 97±15.64[#] | 102±16.9[#] | 95±14.9 | 100±15.9 | 0.003 |
| Mid-thigh | 50±8.56[¶] | 55.8±7.4[¶] | 48±9.56 | 54.1±8.6 | 46±8.95 | 53.2±8.9 | 45.8±7.54 | 52±7.9 | 0.005 |
| Mid Arm | 28.6±6.54[¶] | 32.16±7.6[¶] | 27.5±6.54 | 30±6.5 | 25.6±6.54 | 29.7±6.4 | 22.3±5.64 | 24.6±5.6 | 0.6 |
| Neck | 35.4±5.69[¶] | 38.74±5.6[¶] | 34.56±6.89 | 38.3±3.6 | 33.25±5.45 | 36.2±4.04 | 30.14±3.27 | 32.1±3.1 | 0.0004 |
| **Skinfold thickness (mm)** | | | | | | | | | |
| Biceps | 14±6.54 | 16.8±7.07 | 13.56±6.65 | 15.4±6.7 | 14.52±4.5 | 17.36±5.4 | 14.52±4.65 | 15.2±5.5 | 0.9 |
| Triceps | 22.3±10.25 | 25.0±10.5 | 21.23±8.65 | 24.3±9.33 | 23.54±6.7 | 24.3±7.7 | 22.12±8.65 | 22.2±9.3 | 0.44 |
| Subscapular | 25±8.65 | 30±8.1 | 26.5±9.23 | 29.2±9.8 | 24.52±6.5 | 27±5.6 | 23.56±5.69 | 26±6.5 | 0.2 |
| Antiaxillary | 12±6.54 | 17±6.0 | 11.4±5.64 | 14.6±5.3 | 10.24±3.5 | 13.27±5.2 | 10.24±5.98 | 13.7±5.1 | 0.5 |
| Suprailiac | 29.5±9.87* | 31.8±9.8* | 27.45±9.56 | 29.2±10.5 | 25.46±8.95 | 28.1±9.5 | 23.24±3.75 | 27±8.9 | 0.02 |
| Lateral thoracic | 29.65±10.23[¶] | 33.7±11.1[¶] | 28.65±11.23 | 31.9±12.5 | 30.1±12.45 | 30.1±11.9 | 28.9±10.25 | 28.9±9.8 | 0.02 |
| Thigh | 27.89±10.95[¶, Y] | 30.1±11.3[¶, Y] | 26±9.8 | 26±9.9 | 25.4±7.54 | 25.4±6.6 | 24.7±10.24 | 24.7±8.1 | 0.05 |

Results are shown as mean± SD. P value ≤0.05 is statistically significant. ANCOVA test were carried out.

*Group 1 *vs* 2, 1*vs* 3 and 1 *vs* 4 (p≤0.05)

[¶] group 1 *vs* 4 (p≤0.05)

[Y] group 1 *vs* 3 (p≤0.05)

[@] group 2 *vs* 4 (p≤0.05)

[#] group 3 *vs* 4 (p≤0.05).

**Table 3. Biochemical investigations.**

| Variables | OSA with NAFLD (n = 130) | | OSA without NAFLD (n = 100) | | Without OSA with NAFLD (n = 95) | | Without OSA and without NAFLD (n = 85) | | Overall P value |
|---|---|---|---|---|---|---|---|---|---|
| | Age (20–44) | Age (45–60) | Age (20–44) | Age (45–60) | Age 20–44 | Age (45–60) | Age (20–44) | Age (45–60) | |
| Fasting Blood Glucose (mg/dl) | 103±25.2[¶, Y] | 115±26.5[¶, Y] | 104.1±38.4[‡] | 112±37.6[‡] | 98.14±21.2 | 105.6±22.3 | 96.3±24.4 | 99.62±24.4 | 0.004 |
| Serum Triglycerides (mg/dl) | 158±41.21[*] | 189±40.6[*] | 164±45.98[@] | 177±46.9[@] | 151±54.62[#] | 158.1±55.2[#] | 149±58.69 | 151±58.9 | 0.01 |
| Total Cholesterol (mg/dl) | 185±38.3[*] | 199±37.89[*] | 180±44.6 [@] | 191±41.56 [@] | 178±43.6 [#] | 185.6±42.3 [#] | 171±39.8 | 180.2±35.65 | 0.02 |
| High density lipoprotein (mg/dl) | 41.23±7.56[¶] | 42.4±8.3[¶] | 42.36±10.23 | 43.8±11.7 | 42.65±8.95[#] | 44.6±9.1[#] | 50.23±9.56 | 52.3±10.2 | 0.005 |
| Low density lipoprotein | 110.2±39.56[¶] | 112.6±40.2[¶] | 108.5±356[@] | 109±36.5[@] | 107.5±34.5[#] | 109±35.6[#] | 98±30.8 | 100±29.65 | 0.002 |
| Very low density lipoprotein | 31.23±10.23 | 33.5±11.2 | 31.21±10.23 | 32±12.3 | 30.21 ±10.24 | 31±11.3 | 29.32±10.21 | 30.0±9.6 | 0.4 |
| Aspartate transaminase (IU/L) | 42.31±13.2 [¶, Y] | 44.5±15.9 [¶, Y] | 40.21±21.2[‡] | 41.4±22.1[‡] | 38.54±18.6 [#] | 39.6±19.6 [#] | 30.21±15.42 | 31.6±15.9 | 0.01 |
| Alanine transaminase (IU/L) | 58.65±10.24 [*] | 60.9±10.3 [*] | 56.54±11.2[@] | 54.2±12.9[@] | 51.23 ±10.54 | 52.3±11.9 | 48.65±9.54 | 50.9±10.9 | 0.03 |
| Alkaline phosphate (IU/L) | 235.6±73.2 [¶, Y] | 240.6±74.3 [¶, Y] | 240±72.32 | 242±76.5 | 231.6 ±70.21 | 235±72.9 | 230.1±69.8 | 235±69.8 | 0.05 |
| Fasting Insulin (µU/ml) | 9.23±2.98[*] | 12±4.3[*] | 9.54±4.54[‡] | 11.1±4.8[‡] | 8.56±3.64 | 9.3±3.6 | 7.56±3.8 | 9.37±3.8 | 0.001 |
| HOMA-IR | 1.9±0.912[*] | 2.9±0.92[*] | 2.1±0.96[@] | 2.5±0.98[@] | 1.8±0.83 | 1.9±0.86 | 1.51±0.76 | 1.6±0.76 | 0.001 |

Results are shown as mean± SD. ANCOVA test were carried out. P value ≤0.05 is statistically significant.

[*]Group 1 *vs* 2, 1*vs* 3 and 1 *vs* 4 (p≤0.05)

[‡]group 2 *vs* 3 and group 2 vs 4 (p≤0.05)

[#] group 3 vs 4 (p≤0.05)

[@] group 2 *vs* 4 (p≤0.05)

[¶] group 1 *vs* 4 (p≤0.05)

[Y] group 1 *vs* 3 (p≤0.05). HOMA-IR, homoeostasis modal assessment for insulin resistance.

was obtained in OSA and NAFLD (p = 0.05). The deviation from Hardy-Weinberg equilibrium among OSA and NAFLD patients for *IRS1 (*Gly972Arg) (p = 0.001) indicated significant association between this SNP and the presence of OSA and NAFLD. The overall genotypic

**Table 4. Allele distribution of IRS1 and IRS2 genes polymorphisms between the groups.**

| | OSA with NAFLD (N = 130) | OSA without NAFLD (N = 100) | Without OSA with NAFLD (N = 95) | Without OSA and without NAFLD (N = 85) | P [a] | Odds ratio (95% CI) | P [b] |
|---|---|---|---|---|---|---|---|
| *IRS1* [Gly972Arg, n (%)] | | | | | | | |
| Allele Gly | 112 (86.5) | 89 (89) | 86 (90.5) | 79 (93) | 0.04 | 1 (reference) | 0.002 |
| Allele Arg | 18 (13.5) | 11 (11) | 9 (9.5) | 6 (7) | | 2.25 (1.41–3.30) | |
| Additive model | | | | | | 4.04 (1.52–11.51) | 0.001 |
| *IRS2* [Gly1057Asp, n ((%)] | | | | | | | |
| Allele Gly | 118 (91) | 92 (92) | 88 (93) | 82 (96.5) | 0.06 | 1 (reference) | 0.08 |
| Allele Asp | 12 (9) | 8 (8) | 7 (7) | 3 (3.5) | | 0.98 (0.75–1.99) | |
| Additive model | | | | | | 1.10 (0.8–2.56 | 0.06 |

Results are shown as n (%). P value ≤0.05 is statistically significant.

[a]P value was computed by the Pearson chi-square test.

[b]Data were calculated by logistic regression after adjusting for age, body mass index, fat mass, % body fat, fasting blood glucose, serum triglyceride, total cholesterol and fasting insulin.

frequency of IRS2 (Gly1057Asp) was 86.66% of subjects were *Gly/Gly* homozygous, 10.42% were Gly/Asp heterozygous, and 2.92% were Asp/Asp homozygous (Table 4). The *IRS2* (Gly1057Asp) genotype frequencies did not follow Hardy Weinberg Equilibrium (chi value = 10.5).

## Multivariate logistic regression

Multivariate logistic regression analyses showed the carriers of homozygous *IRS1* Arg had an increased risk of OSA and NAFLD after adjusting for age, body mass index, fat mass, % body fat, FBG, TG, TC, and fasting insulin (OR 4.49, 95% C.I. 1.06–12.52, p = 0.002) (Table 4).

## Comparison of IRS 1 and IRS-2 genotypes with clinical phenotypes

Association of *IRS1* and IRS-2 gene polymorphisms with clinical, body composition, anthropometric and biochemical parameters is shown in S1 and S2 Tables. In OSA and NAFLD group, BMI, fat mass, % body fat, FBG, serum TG, TC, ALT, AST, fasting insulin and HOMA-IR levels were significantly increased in Gly/Arg genotype as compared to Gly/Gly genotype (Fig 1). In OSA without NAFLD group, only BMI (p = 0.01) was significantly increased in Gly/Arg genotype (S1 Table). In group 3 and group 4, we did not find any significant association between the genotypes. *IRS2* gene polymorphism did not find any significant association between all the groups (S2 Table).

## Severity of OSA

Based on the subgroup analysis according to the severity of OSA (Table 5), allele frequencies of *IRS1* [Gly972Arg] were 92% (Gly) and 8% (Arg) in the mild group, and 85% (Gly) and 15% (Arg) in the moderate group, and 76% (Gly) and 24% (Arg) in the severe group with a

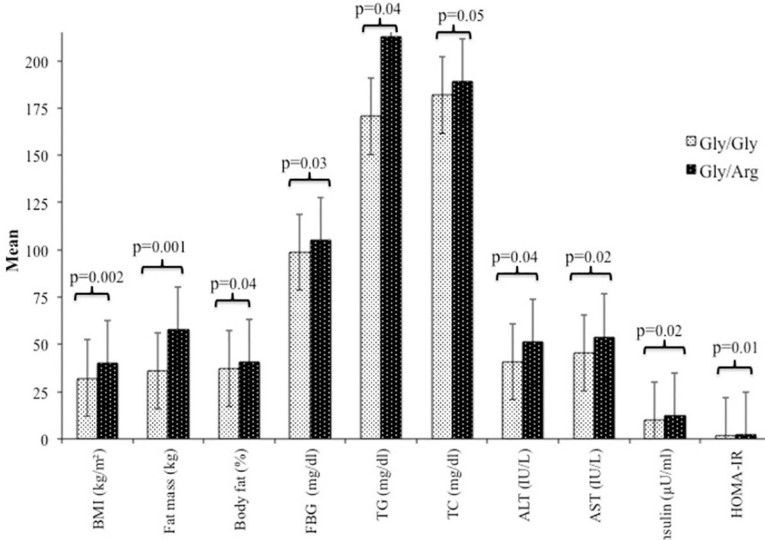

**Fig 1. Association of insulin receptor substrate -1 with clinical, body composition and biochemical parameters in obstructive sleep apnea and non-alcoholic fatty liver disease subjects.** Values are presented in mean and SD. P values<0.05 is statistically significant. BMI, body mass index; FFM, fat free mass; FBG, fasting blood glucose; TG, triglyceride; TC, total cholesterol; ALP, Alkaline phosphate; ALT, alanine transaminase; AST, aspartate transaminase; HOMA-IR, homoeostasis Model Assessment of insulin resistance.

**Table 5. Allele distributions in patients according to the severity of OSA.**

| Gene | | OSA, n (%) | | | Mild vs Moderate | | Mild vs severe | | Moderate vs Severe | |
|---|---|---|---|---|---|---|---|---|---|---|
| | | Mild, | Moderate | Severe | P-value | OR (95% CI) | P-value | OR (95% CI) | P-value | OR (95% CI) |
| | | **120 (52)** | 75 (33) | 35 (15) | | | | | | |
| *IRS1* [Gly972Arg] (n, %) | | | | | | | | | | |
| Alleles | Gly | 110 (92) | 64 (85) | 27 (76) | **0.004** | 8.34(2.56–25.32) | **0.01** | 5.01(1.65–14.62) | **0.05** | 1.23 (0.84–3.32) |
| | Arg | 10 (8) | 11 (15) | 8 (24) | | | | | | |
| *IRS2* [Gly1057Asp] (n, %) | | | | | | | | | | |
| Alleles | Gly | 114 (95) | 67 (90) | 30 (87) | 0.03 | 3.95 (1.65–6.52) | 0.42 | (0.12–0.95) | 0.14 | (0.142–1.61) |
| | Gly | 6 (5) | 8 (10) | 5 (13) | | | | | | |

OSA, obstructive sleep apnea; OR, odds ratio; CI, confidence interval

significant difference (p = 0.004), suggesting that patients with the 'Gly' allele were inclined to develop more severe OSA (p<0.05). We did not find any significant in *IRS2* [Gly1057Asp].

## Discussion

This is the first study to investigate the relationships between *IRS1* and *IRS2* gene polymorphisms with OSA and NAFLD in Asian Indians. In this study, we showed that clinical, body composition, obesity and metabolic parameters were significantly higher in OSA and NAFLD subjects. In addition, higher age group (45–60 years) were significantly increase chances of obesity, hypertension, insulin resistance and T2DM in NAFLD and OSA subjects. Further, this study indicated that the frequency of Arg allele of Gly972Arg polymorphisms of *IRS1* gene is significantly increased in OSA and NAFLD. Importantly, *IRS1* polymorphism is significant genetic determinant for insulin resistance and obesity in OSA and NAFLD. Indeed, subjects carrying IRS1 (Gly/Arg) have significantly higher risk of OSA and NAFLD.

Several cross-sectional studies examined levels of hepatic enzymes in patients with OSA [22–25]. Chin *et al.* [22] reported that elevated fasting AST levels in OSA patients and correlated with insulin resistance. Another study, Norman *et al.* [23] showed that ALT and AST levels were directly correlated with the severity of nocturnal hypoxia. Increased levels of ALT, AST and AP have been indicated in patients with moderate and severe OSA [24]. Sing *et al* *[25]* found OSA was prevalent in 46% of patients with higher AST levels. Similarly, the presence of severe OSA (AHI > 50/ minute) is an independent predictor of elevated liver enzymes [26]. Based on these studies, an interesting finding in our study indicates that metabolic and liver markers are significantly higher in OSA with NAFLD patients.

From developed countries, a limited number of studies related to *IRS1* and *IRS2* gene polymorphisms focused on T2DM, OSA and NAFLD separately, but no study has been investigated the polymorphism in patients with OSA and NAFLD both. Li *et al.* [27] have shown that *IRS1* gene plays an important role in T2DM risk, especially in Asian. It also indicates that IRS1 gene polymorphism is associated with T2DM risk in Caucasian. Another study, Dongiovanni *et al.* [11] reported that *IRS1* (Gly972Arg) polymorphism affects insulin receptor activity and predisposes to liver damage and decreases hepatic insulin signaling in patients with NAFLD. Li et al. [28] recruited 130 patients with obstructive sleep apnea hypopnea syndrome (OSAHS) and 136 age and gender matched healthy controls. He showed allele and genotype frequencies of *IRS1* gene showed significant differences between OSAHS and controls in the Chinese Han population. Our study also showed significant association of the *IRS1* (Gly/Arg) gene with OSA and NAFLD. Further, in a study from Turkey on 972 OSA subjects, the polymorphism of

the *IRS1* (Gly/Arg) was associated with the occurrence of OSAS in male patients, whereas this polymorphism was not related to the severity of OSAS [29], which was inconsistent with our finding. The discrepancy between our results and the previous result could be attributed to ethnicity, environmental factors and probably due to larger sample size of our study compared to previous study.

Insulin resistance is the key factor in NAFLD and OSA pathophysiology as well as in the progression of the disease. IRS1 and IRS2 are important for the development of NAFLD in the presence of insulin resistance. Insulin resistance signaling is an exclusively mediated by IRS1 and IRS2 in the liver [30]. In this context, Mkadem *et al.* [31] suggested that *IRS1* Arg972 alleles are more prevalent in insulin-resistant subjects, and these alleles are also prevalent in overweight/ obese individuals. Another study reported that the effect of *IRS1* polymorphism on hepatic insulin resistance and he showed decreased hepatic levels, reflecting reduced insulin signaling activity [32]. Interestingly, our study also indicated that *IRS1* polymorphism is significant genetic determinant for insulin resistance in OSA and NAFLD.

In the Pima Indians, the frequency of *IRS2* gene polymorphism is the highest compared to other populations [33]. This may be because of the high prevalence of obesity and T2DM in the population. Additionally, the current research we did not find any association of *IRS2* gene with OSA and NAFLD patients. We believe that the *IRS2* (Gly1057Asp) polymorphism influence glucose homeostasis and obesity. A molecular mechanism related to *IRS2* polymorphism is still unknown. Based on these observations, it seems reasonable to speculate that *IRS-2* variants are not involved in the development of OSA and NAFLD.

Limitations of our study include samples are originated from north India. There is also the lack of data on siblings and other ancestral members of the recruited subjects, which could help in determining the effect of population stratification. Another limitation of our study is the lack of biopsy data and other ancestral members of the recruited subjects, which could help in determining expression across populations for the effect of population stratification. Further, although ultrasonography is a practical approach commonly used to detect liver steatosis, it is not the gold standard technique for quantitative liver fat assessment. Further, ultrasonography is the most common procedure for diagnosis of NAFLD in clinical practice and has a fair sensitivity (87%) and specificity (94%) in detecting hepatic steatosis [34]. It is simple to perform, non-invasive, cost-effective and does not entail any radiation hazard, and could also be used in the epidemiological studies.

## Conclusion

Genetic factors may predispose to OSA and NAFLD. We observed significant association of the *IRS1* (Gly/Arg) gene with OSA and NAFLD, whereas *IRS2* (Gly1057Asp) polymorphism is not related to the severity of OSA and NAFLD. Further, *IRS1* polymorphism is a significant genetic determinant for insulin resistance in OSA and NAFLD.

## Supporting information

**S1 Table. Association of IRS-1 gene polymorphism with clinical, body composition, anthropometry and biochemical parameters.**
(DOCX)

**S2 Table. Association of IRS-2 gene polymorphism with clinical, body composition, anthropometry and biochemical parameters.**
(DOCX)

## Acknowledgments

The authors acknowledge the contribution of Mr. Kirti Pratap who performed many of the biochemical investigations. Finally, the cooperation of the subjects who took part in the study is greatly appreciated.

## Author Contributions

**Conceptualization:** Surya Prakash Bhatt.

**Data curation:** Surya Prakash Bhatt.

**Formal analysis:** Surya Prakash Bhatt.

**Funding acquisition:** Surya Prakash Bhatt.

**Investigation:** Surya Prakash Bhatt.

**Methodology:** Surya Prakash Bhatt.

**Project administration:** Surya Prakash Bhatt.

**Resources:** Surya Prakash Bhatt.

**Software:** Surya Prakash Bhatt.

**Supervision:** Surya Prakash Bhatt.

**Validation:** Surya Prakash Bhatt, Randeep Guleria.

**Visualization:** Surya Prakash Bhatt.

**Writing – original draft:** Surya Prakash Bhatt.

**Writing – review & editing:** Surya Prakash Bhatt, Randeep Guleria.

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
