## [Decision Letter · Decision Letter 0]

15 Mar 2021

PONE-D-20-40461

Association of IRS1 (Gly972Arg) and IRS2 (Gly1057Asp) genes polymorphisms with OSA and NAFLD in Asian Indians.

PLOS ONE

Dear Dr. Bhatt,

Thank you for submitting your manuscript to PLOS ONE. After careful consideration, we feel that it has merit but does not fully meet PLOS ONE’s publication criteria as it currently stands. Therefore, we invite you to submit a revised version of the manuscript that addresses the points raised during the review process.

Your manuscript has been received two reviews. There was consistency in the reviews on the lack of clarity in the statistical analysis.  Further. please address each of the comments from the two reviewers. Please note that one of the criteria for publication in PLOS One relates to the need to reduce redundant work.

We look forward to receiving your revised manuscript.

Kind regards,

Michael W Greene, Ph.D.

Academic Editor

PLOS ONE

Reviewers' comments:

Reviewer's Responses to Questions

**Comments to the Author**

1. Is the manuscript technically sound, and do the data support the conclusions?

Reviewer #1: Yes

Reviewer #2: Partly

2. Has the statistical analysis been performed appropriately and rigorously? 

Reviewer #1: Yes

Reviewer #2: No

3. Have the authors made all data underlying the findings in their manuscript fully available?

Reviewer #1: Yes

Reviewer #2: Yes

4. Is the manuscript presented in an intelligible fashion and written in standard English?

Reviewer #1: Yes

Reviewer #2: Yes

5. Review Comments to the Author

Reviewer #1: The auhtors evaluated the association of IRS1 gene polymorphism with sleep apnea and hepatosteatosis in a specific poluplation, and found an association of that polymorphism with sleep apnea and fatty liver.

They performed a huge work with numerous statistical comparisons in the text and tables, which are not easy to follow for a raeder.

Practically, the majority of these comparisons already exist in the literature, and clinicans are familiar with these risk factors associated with sleep apnea and fatty liver. Therefore, the conlusions of the study are not quite original.

I would expect to see whether there was a correlation between the severity of sleep apnea and polymorphism, which would make sense in the clinical practise.

Specific comment: Discussion, line 248, "turkey" should be wirteen as "Turkey", their meanings are quite different!

Reviewer #2: The author's Bhatt and Guleria describes the relationship between IRS1 and IRS2 with obstructive sleep apnea (OSA) and non-alcoholic fatty liver disease (NAFLD) in 410 overweight/obese subjects age ranging from 20-60 years. The results discuss the difference between OSA with or without NAFLD and NAFLD without OSA among the participant's clinical/ biochemical investigations and anthropometric measurements. They showed the high risk of OSA and NAFLD with IRS1 and not IRS2 as compared. This article brings some additional data to this field of research. I have few concerns that need to be addressed.

1. Introductions need to be updated with recent publications.

2. The study was conducted with 20-60 years old participants. Why the data analysis is done as a single age group 20-60 years, participants aged above 45 will have more chance of risk factor for type 2 diabetes. Since the manuscript deals with the primary risk factor for NAFLD is due to obesity and type 2 diabetes, separating the age group in the study is essential.

3. In the methodology, the statistical section needs to be explained more clearly. The authors need to mention each table calculated using either ANCOVA or ANOVA.

4. As the authors' claims, the statistical significance is not showing significant compared to the symbol mentioned groups in Table 1 and 2.

For example, in Table 1

Diastolic pressure did not show any significance than any group by ANOVA using Bonferroni post-hoc test (Using GraphPad prism statistical analysis with the table's data).

Systolic pressure # - Not significant

BMI # -Not significant

body fat U,@, # - Not significant

Table 2

Waist - @

Hip - @, #

5. The authors need to check the statistical analysis thoroughly since their hypothesis fully depends upon the statistical analysis.

6. PLOS authors have the option to publish the peer review history of their article (what does this mean?). If published, this will include your full peer review and any attached files.

Reviewer #1: **Yes: **Yıldırım A. Bayazıt

Reviewer #2: No

---

## [Author Response · Author response to Decision Letter 0]

3 Jul 2021

Response to Reviewers’ comments:

The authors are thankful to the reviewers for their valuable comments and suggested modifications. The responses to the comments of the reviewers are presented below. 

Reviewer #1: 

General Comment: The authors evaluated the association of IRS1 gene polymorphism with sleep apnea and hepatosteatosis in a specific population, and found an association of that polymorphism with sleep apnea and fatty liver.

Reply: Thank you so much for this comment

Comment 1: They performed a huge work with numerous statistical comparisons in the text and tables, which are not easy to follow for a reader.

Reply 1: We greatly appreciate the comment. According to reviewer comment, we have reanalysed the data and modified accordingly. 

Comment 2: Practically, the majority of these comparisons already exist in the literature, and clinicians are familiar with these risk factors associated with sleep apnea and fatty liver. Therefore, the conclusions of the study are not quite original.

Reply 3: We thank reviewer for pointing out this issue. We know these polymorphisms were investigated in other population and diseases but not in Asian Indian population particularly combined in OSA and NAFLD. However, we hypothesized that the IRS1 (Gly972Arg) and IRS2 (Gly1057Asp) genes may influence insulin resistance and are associated with risk of OSA and NAFLD in overweight non-diabetic Asian Indians. This is the first study to investigate the relationships between IRS1 and IRS2 gene polymorphisms with OSA and NAFLD in Asian Indians.

Comment 3: I would expect to see whether there was a correlation between the severity of sleep apnea and polymorphism, which would make sense in the clinical practise.

Reply: We greatly appreciate the comment. We have reanalysed the complete data and modifications has been incorporated. 

“Based on the subgroup analysis according to the severity of OSA (table 5), allele frequencies of IRS1 [Gly972Arg] were 92% (Gly) and 8% (Arg) in the mild group, and 85% (Gly) and 15% (Arg) in the moderate group, and 76% (Gly) and 24% (Arg) in the severe group with a significant difference (p=0.004), suggesting that patients with the ‘Gly’ allele were inclined to develop more severe OSA (p<0.05). We did not find any significant in IRS2 [Gly1057Asp].” 

Comment 4: Specific comment: Discussion, line 248, "turkey" should be written as "Turkey", their meanings are quite different!

Reply 3: We thank reviewer for pointing out this issue. We have re- checked the manuscript and the errors has been corrected.

Reviewer #2: 

General Comment: The author's Bhatt and Guleria describes the relationship between IRS1 and IRS2 with obstructive sleep apnea (OSA) and non-alcoholic fatty liver disease (NAFLD) in 410 overweight/obese subjects age ranging from 20-60 years. The results discuss the difference between OSA with or without NAFLD and NAFLD without OSA among the participant's clinical/ biochemical investigations and anthropometric measurements. They showed the high risk of OSA and NAFLD with IRS1 and not IRS2 as compared. This article brings some additional data to this field of research. I have few concerns that need to be addressed.

Rely: Thank you so much for this comment

Comment 1: Introductions need to be updated with recent publications.

Reply 3: We thank reviewer for pointing out this issue. We have modified the introduction part. 

“Obstructive sleep apnea (OSA) is a common sleep problem in which complete airway obstruction, caused by pharyngeal collapse during sleeping time. The global prevalence in general populations is 9-38% (1). In Indian studies, the prevalence of OSA is 4.4-13.7% (2). In addition, OSA in Indian males varies from 4.4-19.7% and in females, it is between 2.5- 7.4% from the previous studies (2). The prevalence of OSA also varies depending on the diagnostic criteria used and the age and sex of the population.

Non-alcoholic fatty liver disease (NAFLD) has a broad spectrum from fatty infiltration to severe fibrosis, cirrhosis, and hepatocellular carcinoma. The global prevalence of NAFLD among general population ranged from 11.2% to 37.2 %). Similar ranges were shown of biopsy-confirmed nonalcoholic steatohepatitis (NASH) among NAFLD patients ranging from 15.9% to 68.3% (3). Prevalence of NAFLD in the Asian population was reported as 31%. (4). In our previous study, we reported that the prevalence of NAFLD is 24.5-32.2% (5) and the primary risk factors for NAFLD are obesity, type 2 diabetes mellitus (T2DM), dyslipidemia, and insulin resistance.

Clinical finding showed that OSA has been associated with NAFLD (6). Recent met analysis has been reported that OSA was independently associated with NAFLD in terms of liver enzymes and histological alterations (6). OSA causes accumulation of fatty acids in the liver as a result of nocturnal hypoxia, insulin resistance, metabolic syndrome, dyslipidemia, hypertension, oxidative stress and systemic inflammation. Another study has been indicated that nocturnal hypoxia causes NAFLD development and progression (7). However, nocturnal hypoxia is correlated with development and progression of NAFLD in OSA patients (8). 

Insulin receptor is a hetero tetramer consisting of alpha (α) and beta (β) dimers. The α-subunit consisting of the ligand-binding site, while the β-subunit consists of a ligand-activated tyrosine kinase. On ligand binding, when tyrosine is phosphorylated, the insulin receptor gets converted into two intracellular substrates, insulin receptor substrate (IRS)-1 and insulin receptor substrate (IRS)-2 (9). The gene for IRS1 is located on chromosome 2q36 and encodes a 1,242-amino acid protein. The most common polymorphism in the IRS -1 gene (Gly972Arg), was reported to be associated with OSA (10) and NAFLD (11). The IRS2 gene is located on chromosome 13q34 and encodes a protein of 1,354 amino acids. Moreover, the common polymorphism Gly1057Asp in the IRS2 gene has also been reported to influence the susceptibility to insulin resistance and T2DM in polycystic ovary syndrome women (13). Till date, no studies have been investigated the association of IRS1 and IRS2 gene with OSA and NAFLD in Asian Indians.

 We hypothesized that the IRS1 (Gly972Arg) and IRS2 (Gly1057Asp) genes may influence insulin resistance and are associated with risk of OSA and NAFLD in overweight non-diabetic Asian Indians. The aim of the present study was to investigate the relationships between IRS1 and IRS2 gene polymorphisms with OSA and NAFLD in Asian Indians.”

References:

1. Chamara V Senaratna , Jennifer L Perret , Caroline J Lodge , Adrian J Lowe , Brittany E Campbell , Melanie C Matheson et al. Prevalence of obstructive sleep apnea in the general population: A systematic review. Sleep Med Rev. 2017 Aug; 34:70-81. 

2. Reddy EV, T Kadhiravan, HK Mishra, V Sreenivas, KK Handa, S Sinha, et al. 2009. Prevalence and risk factors of obstructive sleep apnea among middle-aged urban Indians: a community-based study. Sleep Med, 10(8): 913e8.

3. Jean-FrançoisDufour, RogerScherera, Maria-MagdalenaBalp, et al. The global epidemiology of nonalcoholic steatohepatitis (NASH) and associated risk factors–A targeted literature review. Endocrine and Metabolic Science. Volume 3, 30 June 2021, 100089

4. J. Li, B. Zou, H. Fujii, Y.H. Yeo, F. Ji, D.H. Lee, et al. Prevalence of non-alcoholic fatty liver disease (NAFLD) in Asia: a systematic review and meta-analysis of 195 studies and 1,753,168 subjects from 15 countries and areas. Gastroenterology, 154 (2018) S-1165.

5. Bajaj S, A Nigam, A Luthra, RM Pandey, D Kondal, SP Bhatt, et al. 2009. A case-control study on insulin resistance, metabolic co-variates & prediction score in non-alcoholic fatty liver disease. Indian J Med Res. 129:285-92

6. Jin S, Jiang S, Hu A. Association between obstructive sleep apnea and non-alcoholic fatty liver disease: a systematic review and meta-analysis. Sleep Breath. 2018. Jan 15. doi: 10.1007/s11325-018-1625-7.

7. Mirrakhimov AE, VY Polotsky. 2012. Obstructive sleep apnea, and non-alcoholic Fatty liver disease: is the liver another target? Front Neurol. 17; 3(): 149.

8. Erol Cakmak, Faysal Duksal,Engin Altinkaya, Fettah Acibucu, Omer Tamer Dogan, Ozlem Yonem, and Abdulkerim Yilmaz1 Association Between the Severity of Nocturnal Hypoxia in Obstructive Sleep Apnea and Non-Alcoholic Fatty Liver Damage. Hepat Mon. 2015 Nov; 15(11): e32655. 

9. Pierre De Meyts. The Insulin Receptor and Its Signal Transduction Network. 2016 

10. Bayazit YA, ME Erdal, M Yilmaz, TU Ciftci, F Soyleme Z, T Gokdoğan, O Kokturk, YK Kemaloglu, A Koybasioglu. 2006. Insulin receptor substrate gene polymorphism is associated with obstructive sleep apnea syndrome in men. Laryngoscope. 116(11): 1962-5.

11. Dongiovanni P, L Valenti, R Rametta, AK Daly, V Nobili, E Mozzi, et al. 2010. Genetic variants regulating insulin receptor signalling are associated with the severity of liver damage in patients with non-alcoholic fatty liver disease. Gut. 59(2): 267

12. Villuendas G, JI Botella-Carretero, B Roldán, J Sancho, HF Escobar-Morreale, JL San Millán. 2005. Polymorphisms in the insulin receptor substrate-1 (IRS-1) gene and the insulin receptor substrate-2 (IRS-2) gene influence glucose homeostasis and body mass index in women with polycystic ovary syndrome and non-hyperandrogenic controls. Hum Reprod. 20(11): 3184-91

13. Dudeja V, A Misra, RM Pandey, et al. 2001. BMI does not accurately predict overweight in Asian Indians in Northern India. Br J Nutr. 86: 105-112.

Comment 2: The study was conducted with 20-60 years old participants. Why the data analysis is done as a single age group 20-60 years, participants aged above 45 will have more chance of risk factor for type 2 diabetes. Since the manuscript deals with the primary risk factor for NAFLD is due to obesity and type 2 diabetes, separating the age group in the study is essential.

Reply 3: We thank reviewer for pointing out. As suggested by reviewer, we re-analysed the data and the tables and results has been charged. 

Comment 3: In the methodology, the statistical section needs to be explained more clearly. The authors need to mention each table calculated using either ANCOVA or ANOVA.

Reply 3: We thank reviewer for pointing out this issue. The errors has been corrected

“Data was entered in an Excel spreadsheet (Microsoft Corp, Washington, USA). The distribution of clinical, biochemical, anthropometric and body composition parameters were confirmed to approximate normality. Categorical data was analysed by Chi-squared test, with Fisher correction when appropriate, and expressed as absolute number (%). Continuous variables were expressed as the mean ± standard deviation to summarize the variables. All continuous values were performed using the Z score method. The influence of the groups (1vs2, 1 vs3, 1vs 4, 2vs3 and 2vs4) was estimated by the Analysis of Covariance (ANCOVA) test with multiple comparisons. Pearson’s correlation coefficient and significance of ‘r’ were used to compare the inflammatory marker levels and clinical parameters.

In order to determine if observed allele frequency was in conformity with the expected frequency (Hardy Weinberg equilibrium), chi-square analysis was done. Between-group differences in proportions of alleles or genotypes were compared using Chi-square test and a two-tailed Fisher’s exact test. The influence of the genotype on the clinical biochemical, anthropometric and body composition parameters was estimated by ANCOVA, whether there are any statistically significant differences between the groups. Logistic regression analyses were carried out to identify the differences in genotypic frequencies and interaction of two SNPs between the groups. Bonferroni corrections for multiple comparisons were performed. The odds ratio (OR) and 95% confidence interval were used as a measure of strength for the association between IRS1 (Gly972Arg) and IRS2 (Gly1057Asp) genotypic combinations with the disease. In addition, subgroup analysis was conducted to see the relationship of OSA and gene polymorphisms. A p-value <0.05 was considered as significant.”

Comment 4: As the authors' claims, the statistical significance is not showing significant compared to the symbol mentioned groups in Table 1 and 2.

For example, in Table 1

Diastolic pressure did not show any significance than any group by ANOVA using Bonferroni post-hoc test (Using GraphPad prism statistical analysis with the table's data).

Systolic pressure # - Not significant

BMI # -Not significant

body fat U,@, # - Not significant

Table 2

Waist - @

Hip - @, #

Reply 5: We thank reviewer for pointing out this issue. We have re-analysed the data and modified accordingly.

Comment 5: The authors need to check the statistical analysis thoroughly since their hypothesis fully depends upon the statistical analysis.

Reply: We greatly appreciate the comment. Please see same as reply 3

Comment 5: English spelling and grammar should be thoroughly reviewed by the authors.

Reply: Thank you so much for this comment. The errors has been corrected.

---

## [Decision Letter · Decision Letter 1]

2 Aug 2021

Association of IRS1 (Gly972Arg) and IRS2 (Gly1057Asp) genes polymorphisms with OSA and NAFLD in Asian Indians.

PONE-D-20-40461R1

Dear Dr. Bhatt,

We’re pleased to inform you that your manuscript has been judged scientifically suitable for publication and will be formally accepted for publication once it meets all outstanding technical requirements.

Kind regards,

Michael W Greene, Ph.D.

Academic Editor

PLOS ONE

Reviewers' comments:

Reviewer's Responses to Questions

**Comments to the Author**

1. If the authors have adequately addressed your comments raised in a previous round of review and you feel that this manuscript is now acceptable for publication, you may indicate that here to bypass the “Comments to the Author” section, enter your conflict of interest statement in the “Confidential to Editor” section, and submit your "Accept" recommendation.

Reviewer #1: All comments have been addressed

Reviewer #2: All comments have been addressed

2. Is the manuscript technically sound, and do the data support the conclusions?

Reviewer #1: Yes

Reviewer #2: Yes

3. Has the statistical analysis been performed appropriately and rigorously? 

Reviewer #1: Yes

Reviewer #2: Yes

4. Have the authors made all data underlying the findings in their manuscript fully available?

Reviewer #1: Yes

Reviewer #2: Yes

5. Is the manuscript presented in an intelligible fashion and written in standard English?

Reviewer #1: Yes

Reviewer #2: Yes

6. Review Comments to the Author

Reviewer #1: The authors revised the statisttics of the manuscript for the readers to follow.

Regarding the originality of the study, they also hypothesized that the IRS1 (Gly972Arg) and IRS2 (Gly1057Asp) genes may influence insulin resistance and are associated with risk of sleep apnea in overweight non-diabetic Asian Indians.

Regarding the clinical value of the manuscript, they reanalysed the data

and, added table 5.

They also corrected the typographical errors

Reviewer #2: The authors carried out the reviewer's comments and modified the manuscript text and tables. The revised results are incorporated in the result section.

7. PLOS authors have the option to publish the peer review history of their article (what does this mean?). If published, this will include your full peer review and any attached files.

Reviewer #1: No

Reviewer #2: No

---

## [Editor Report · Acceptance letter]

19 Aug 2021

PONE-D-20-40461R1 

Association of *IRS1* (Gly972Arg) and *IRS2* (Gly1057Asp) genes polymorphisms with OSA and NAFLD in Asian Indians. 

Dear Dr. Bhatt:

I'm pleased to inform you that your manuscript has been deemed suitable for publication in PLOS ONE. Congratulations! Your manuscript is now with our production department. 

Kind regards, 

on behalf of

Dr. Michael W Greene 

Academic Editor

PLOS ONE